# Assessing the climate change exposure of foreign direct investment

Xia Li [1✉] & Kevin P. Gallagher [2✉]

This study deploys newly available data to examine the exposure of multinational companies' overseas investments to physical climate risks. Globally, foreign investments are significantly exposed to lower physical climate risks, compared with local firms across countries. Within countries however, the differences of physical climate risks between foreign and local facilities are small. We also examine China, as it is fast becoming one of the largest sources of outward foreign investment across the globe. We find that foreign direct investment from China is significantly more exposed to water stress, floods, hurricanes and typhoon risks across countries, compared with other foreign facilities. Within host countries however, once again the physical climate risks of Chinese overseas facilities are comparable to those of non-Chinese foreign investments.

[1] Questrom School of Business, Global Development Policy Center, Boston University, Boston, MA 02215, USA. [2] Pardee School of Global Studies, Global Development Policy Center, Boston University, Boston, MA 02215, USA. ✉email: xiali7@bu.edu; kpg@bu.edu

Physical climate risks, defined as risks arising from the physical effects of climate change, increasingly affect facilities worldwide across industries[1–6], including foreign assets, or foreign direct investment (FDI)[7]. For instance, increased rainfall and flooding interrupted business at Toyota's manufacturing facilities in Southeast Asia[8]. Water shortage shut down a Coca-Cola plant in India[9]. Risks from rising sea levels affects some of Chinese infrastructure investments in Pakistan[10].

Despite the increasing impact of physical climate risks on firms and facilities globally, little is known about how multinational companies incorporate such risks into their overseas investment decisions. Previous literature related to FDI and the environment focused on the theory of externalities, such as the extent to which firms might locate to countries that have less stringent regulation that requires firms to internalize environmental externalities[11,12]; how foreign companies may spread cleaner environmental technologies or practices in host countries;[13] or whether foreign firms have better environmental performances than indigenous firms[14]. With respect to climate change, studies have primarily focused on the relationship between FDI and carbon emissions[15,16]. While the emerging literature on physical climate risk pays attention to the financial impact of climate change on firm performance, cost of capital, and asset value or price in general[17–20], little attention has been paid to physical climate risks and FDI.

This paper represents an initial foray into this neglected research area and examines the physical climate risks of FDI. In this study, we find that FDI is exposed to lower physical climate risks, compared with local firms across countries. Within host countries however, the differences of physical climate risks between overseas and local facilities are small. We also find that Chinese FDI is exposed to higher climate risk than non-Chinese FDI. Chinese FDI is exposed to higher water stress, floods, and hurricanes and typhoon risks across host countries, compared with non-Chinese overseas facilities. Within host countries, however, the physical climate risks of Chinese overseas facilities are comparable to those of non-Chinese FDI.

## Results

**Incorporating physical climate risks into overseas investment decisions**. Foreign firms tend to shy away from countries with the higher levels of physical climate risks than do local firms (firms that are not multinational or multinationals operating in their headquarter country), which by their nature have less choice regarding where they can locate their facilities. When firms to locate in particular countries, they take on similar levels of risk as do local firms. Chinese FDI on the other hand, is significantly more exposed to most physical climate risks than non-Chinese FDI across countries, but also is not significantly more exposed to such risks within the countries they choose to locate.

We begin by examining the physical climate risks of multinational companies' overseas facilities across the globe. Firms considering locations in areas that are susceptible to physical climate risks will have to bear the costs of climate-related events if they occur. Firms' decisions to locate facilities abroad involves considerations of the characteristics of the host country (e.g., market attractiveness and inputs factors)[21,22] and the firms' own capabilities[23–25]. Compared with local firms, foreign firms investing abroad are at disadvantage in a local market because they lack information about local conditions, face discrimination by host country stakeholders, and have difficulty in responding to some local conditions[26]. To overcome the burden of foreignness and enhance their long-term competitiveness, foreign firms may be more cautious about risks in host countries, including climate risks. It is therefore possible that facilities owned by foreign firms have, on average, lower physical climate risks than those owned by local firms.

We compare whether facilities owned or operated by foreign companies are different from local firms by estimating a set of fixed-effects cross-sectional models based on our firm-host country-industry level climate risk dataset. We find that across host countries, facilities owned by foreign companies have significantly lower climate risks, particularly for floods, seas level rise, and hurricanes/typhoons risks. Within host countries, however, the differences are small and vary among different climate risk drivers. Also, we find that the climate risks of firm's overseas facilities vary by industry, with agriculture and mining industries having the highest aggregate climate risks. In addition, overseas facilities in the Caribbean, the Middle East, and Southeast Asia have the highest climate risks.

We then focus on the physical climate risks of Chinese overseas facilities and examine whether they are different from those of the non-Chinese overseas facilities. China is now among the largest outward foreign investors globally[27,28]. Also, some Chinese overseas investments have political and strategic considerations (e.g., those under the Belt and Road Initiative umbrella) and are not solely profit-seeking[29,30]. They may be more likely to locate in countries with higher risks (including climate risks) if these investments fit with the government's strategy. Further, because Chinese firms have expanded their overseas footprints only recently, they may have had to invest in locations with higher physical climate risks because the less-risky ones have already been taken[31–33].

Descriptive statistics suggest that overseas facilities owned or operated by mainland China and Hong Kong firms have higher aggregate climate risks across countries and industries, compared to overseas facilities owned or operated by companies headquartered in other countries with high FDI outflow stock. Further, we estimate a set of fixed-effects cross-sectional models based on our firm-host country-industry level climate risk data set. We find that overseas facilities owned or operated by Chinese companies have higher water stress, flood, and hurricane/typhoon risks across countries, compared to non-Chinese overseas facilities. Within host countries, however, the climate risks of Chinese overseas facilities are comparable to those of other FDI facilities. We also explore several potential mechanisms explaining why Chinese overseas facilities have higher climate risks across host countries.

Note that physical climate risks are different from carbon risks or transition climate risks - that is, risks arising from transition to a low carbon economy that affect a firms' business[34,35]. A facility's physical climate risks are mainly determined by the facility's location and the nature of its activities. A facility's carbon risks are mainly determined by its energy use, technology choice, and a country's carbon policy. In this paper we focus on physical climate risks and the term "climate risks" refers to physical climate risks unless otherwise specified. Also, we use the term "country" and "jurisdiction" interchangeably. Figure 1 presents the structure of the paper and explains key terminologies.

**Global landscape of climate risks of public companies' overseas facilities**. We compare climate risks of facilities owned or operated by foreign multinational companies with all facilities in the sample. The Methods section details the model specifications (Eqs. (1a) and (1b)) and explains the selection of control variables. We estimate Eq. (1a) (Model 1) to examine whether climate risks of foreign facilities are different from those of all facilities within industry and across host countries, and estimate Eq. (1b) (Model 2) to examine whether climate risks of overseas facilities are different from those of all facilities within industry and within host country. Outcome variables are physical climate risk scores

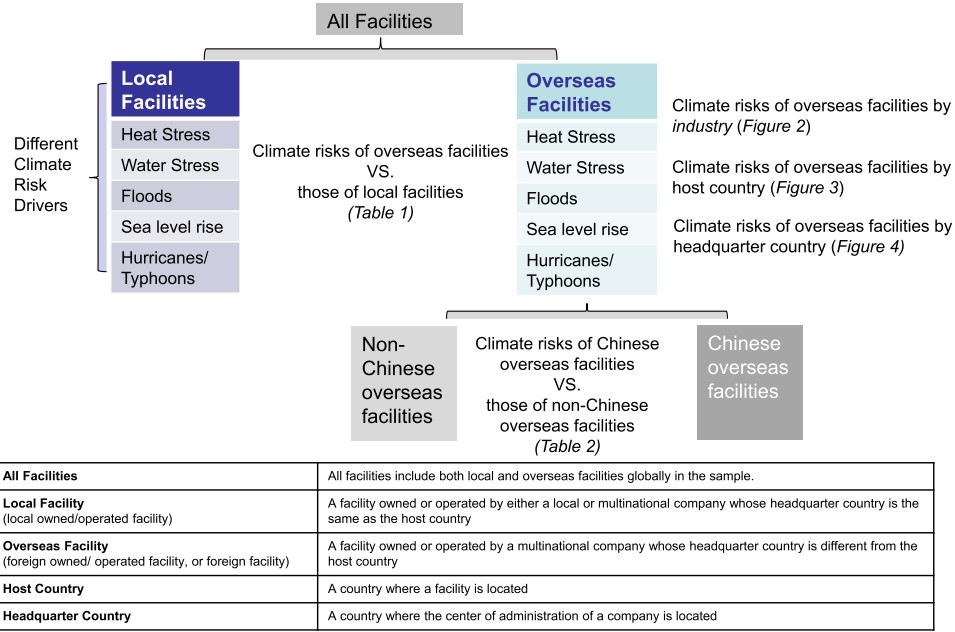

**Fig. 1 Paper structure and terminology.** Presents the structure of the study and explains key terminologies used in the paper.

for different climate risk drivers including heat stress, water stress, floods, sea level rise, and hurricanes/typhoons. The explanatory variable *Foreign* is a dummy which equals to 1 if facilities are owned or operated by foreign companies.

As suggested in Table 1, foreign-owned/operated have lower climate risks across host countries. Specifically, they have significantly and substantially lower floods, seas level rise, and hurricanes/typhoons risks across host countries, compared with local facilities. This is probably because firms are more concerned with host country risks, including climate risks, when investing abroad. They may face discrimination by host country stakeholders, receive more attention because they look different, and have difficulty in responding to some local conditions[14,27]. Within host countries, however, we don't find substantial differences between climate risks of foreign-owned/operated facilities and those of local facilities. Although the differences for some climate risk drivers, such as heat stress, water stress, and floods risk, are statistically significant, they are economically small (e.g., foreign ownership is associated with less than a 2 percent standard deviation difference in heat stress). Also, there is variation amongst climate risk drivers: foreign facilities have higher water stress risk and lower heat stress and floods risk, compared with those of local facilities within host countries. This makes sense, as the climate risks of facilities, whether owned by local or foreign companies, are determined by their locations and the nature of their economic activities and are greatly influenced by the host country's climate. Foreign companies may be less likely to invest in countries with higher climate risks, but if they do, the climate risks they face are likely to be similar to the risks of local companies.

Figure 2 shows the climate risk scores of firms' overseas facilities by industry according to the SIC groups. On average, agriculture and mining industries have the highest aggregate climate risk, while the public administration sector has the lowest climate risk. Specifically, the agriculture, forestry, and fishing industry has the highest heat stress risk; the manufacturing industry has the highest water stress; the mining industry has the highest floods risk; and the whole trade industry has the highest sea level rise and hurricane/typhoon risks. These findings make sense as location-specific assets that are resource-intensive sectors

such as agriculture, mining, and manufacturing with dependent upon natural resources for inputs are more directly affected by chronic risks[36] such as heat and water stresses, while trade and transportation sectors are more directly affected by sea level rise and hurricane/typhoon risks, as their assets are usually near seaports.

Figure 3 compares average climate risk scores of overseas facilities in different countries. The descriptive statistics suggest that overseas facilities in the Caribbean (e.g., Trinidad and Tobago), the Middle East (e.g., Bahrain), and Southeast Asia (e.g., the Philippines) have the highest climate risks. Facilities in Africa (e.g., Rwanda), West Asia (e.g., Saudi Arabia), and South America (e.g., Venezuela) have high heat stress. Facilities in the Middle East (e.g., Bahrain) and central Asia (e.g., Tajikistan and Pakistan) have high water stress. Facilities in Southeast Asia (e.g., Indonesia and Laos) and Central Asia (e.g., Kyrgyzstan) have high floods risk. Facilities on certain islands (e.g., the Faroe Islands and the Solomon Islands) have high sea level rise risk. Facilities in East Asia (e.g., Taiwan, Hong Kong SAR, and Japan) have high hurricane and typhoon risk. Supplementary Fig. 1 in the Supplementary Document summarizes climate risk scores of overseas facilities in the 15 jurisdictions with the highest FDI inflow stock between 1970 and 2019; among these jurisdictions, overseas facilities in Hong Kong SAR have the highest aggregated climate risk.

**Climate risks of Chinese overseas facilities**. Figure 4 summarizes climate risk scores of overseas facilities owned by firms in the 15 jurisdictions with the highest FDI outflow stock between 1970 and 2019. Among those jurisdictions, facilities owned or operated by firms headquartered in China have the highest climate risks across industries and host countries among all foreign operating multinationals. Overseas facilities owned by Hong Kong SAR firms have the highest water stress and floods risks, while facilities owned by mainland Chinese firms have the highest hurricanes/typhoons and sea level rise risks.

The descriptive statistics above suggest that overseas facilities owned or operated by Chinese companies (including Hong Kong SAR) have the highest aggregate climate risks across host

**Table 1 Difference of climate risks of foreign-owned/operated facilities.**

| | Model 1 - across country | | | | | Model 2 - within country | | | | |
|---|---|---|---|---|---|---|---|---|---|---|
| | Heat | Water | Floods | Sea level rise | Hurricanes/Typhoons | Heat | Water | Floods | Sea level rise | Hurricanes/Typhoons |
| Foreign | −0.023 | −0.014 | −0.244 | −0.238 | −0.569 | −0.019 | 0.036 | −0.059 | −0.046 | −0.013 |
| | [0.059] | [0.026] | [0.024]*** | [0.066]*** | [0.043]*** | [0.007]** | [0.013]** | [0.012]*** | [0.033] | [0.009] |
| Controls | | | | | | | | | | |
| Cash | −0.449 | −0.415 | 0.515 | 0.480 | −0.369 | −0.079 | 0.345 | −0.018 | 0.299 | −0.077 |
| | [0.353] | [0.365] | [0.120]*** | [0.348] | [0.436] | [0.034]** | [0.169]* | [0.022] | [0.150]* | [0.066] |
| Size | 0.034 | 0.056 | −0.017 | −0.024 | −0.053 | −0.006 | −0.004 | 0.008 | 0.041 | 0.004 |
| | [0.009]*** | [0.019]** | [0.008]* | [0.009]** | [0.021]** | [0.006] | [0.012] | [0.006] | [0.006]*** | [0.003] |
| ROA | 1.785 | 1.421 | −1.216 | −1.457 | −0.936 | −0.103 | −0.299 | −0.084 | −0.057 | −0.013 |
| | [0.168]*** | [0.328]*** | [0.212]*** | [0.140]*** | [0.275]*** | [0.196] | [0.208] | [0.121] | [0.146] | [0.040] |
| Leverage | 0.096 | −0.008 | 0.002 | 0.019 | −0.186 | 0.019 | 0.140 | 0.016 | 0.016 | 0.011 |
| | [0.190] | [0.110] | [0.119] | [0.116] | [0.247] | [0.026] | [0.030]*** | [0.018] | [0.017] | [0.016] |
| Host country FE | N | N | N | N | N | Y | Y | Y | Y | Y |
| Industry FE | Y | Y | Y | Y | Y | Y | Y | Y | Y | Y |
| N | 51084 | 50665 | 50196 | 51084 | 51084 | 51084 | 50665 | 50196 | 51084 | 51084 |
| r2 | 0.071 | 0.143 | 0.191 | 0.083 | 0.155 | 0.953 | 0.764 | 0.557 | 0.522 | 0.928 |

The unit of analysis is firm-host country-industry. Standard errors are clustered at the industry level. Outcome variables are climate risk scores and are standardized to a mean of 0 and a standard deviation of 1.
***$P < 0.01$; **$P < 0.05$; *$P < 0.1$.

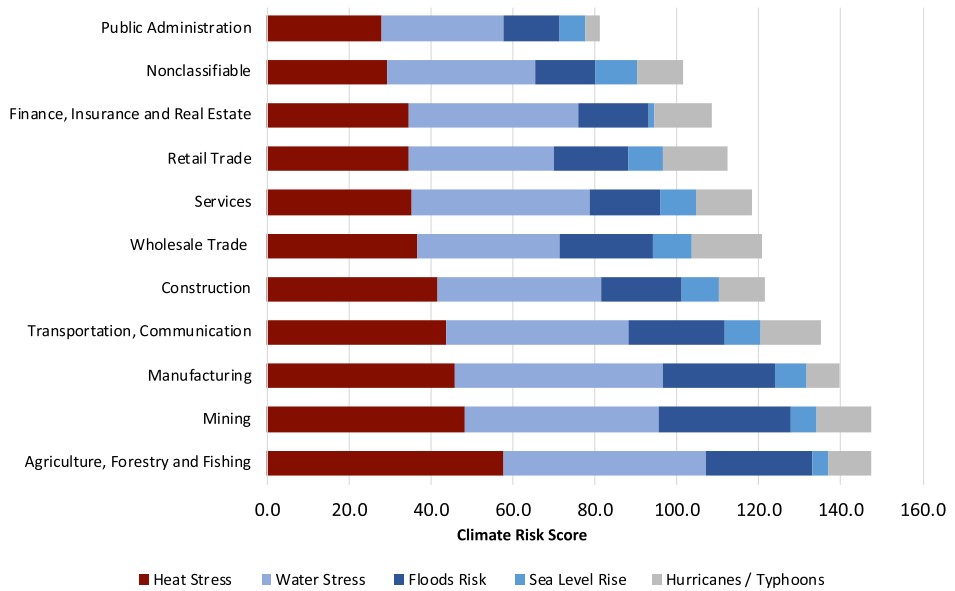

**Fig. 2 Average climate risk scores of overseas facilities by industry.** Analysis is based on climate risk scores and facility statistics of 2233 public companies from Four Twenty Seven. Transportation and Communication sector includes transportation, communications, electric, gas and sanitary service.

countries and industries. However, it is not clear whether the difference is statistically significant, considering industry factors and firm characteristics. We therefore estimate a baseline specification to analyze whether the climate risks of Chinese-owned/operated overseas facilities differ from the average of all FDI facilities. We estimate Eq. (2a) (Model 1) to examine whether climate risks of Chinese-owned/operated overseas facilities differ from those of the global FDI within industry and across host countries, and estimate Eq. (2b) (Model 2) to examine whether climate risks of Chinese-owned/operated facilities differ from the global FDI within industry and host country. Outcome variables are physical climate risk scores for different climate risk drivers: heat stress, water stress, floods, sea level rise, and hurricanes/typhoons. The explanatory variable *ChineseFDI* is a dummy which equals to 1 if overseas facilities are owned or operated by Chinese companies. Each analysis controls for headquarter countries' economic development and carbon emissions and for a set of firm-level control variables. The Methods section details the model specifications (Eqs. (2a) and (2b)) and explains the selection of control variables.

Table 2 presents the results. The statistically significant positive coefficients on *ChineseFDI* in Model 1 suggest that Chinese overseas facilities are exposed to higher water stress, flood, and hurricanes/typhoons risks across host countries (p-values <0.05), compared to all other overseas facilities. The heat stress and sea level rise risks of Chinese overseas facilities do not differ statistically from those of non-Chinese FDI across countries. Results in Model 2 suggest that within a host country, the climate risks of Chinese-owned/operated facilities do not differ significantly from those of non-Chinese overseas facilities except for water stress. Chinese overseas assets are associated with a 9 percent standard deviation decrease in water risk scores within host country (p-values <0.05). The results imply that the higher climate risks of Chinese overseas assets across host countries are driven by the countries Chinese companies invest. In other words, relative to other global public companies, Chinese companies locate facilities in host countries with higher climate risks. Within each host country and industry, Chinese facilities do not tend to be in areas with higher climate risks than are non-Chinese foreign facilities.

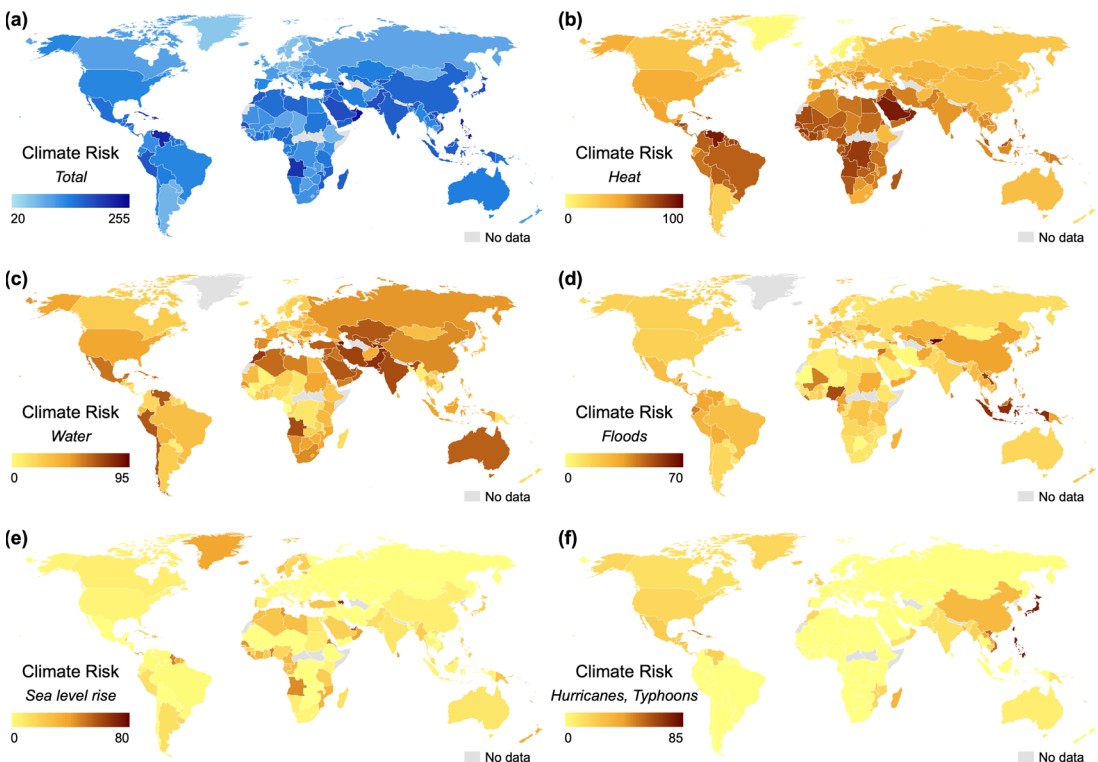

**Fig. 3 Average climate risk scores of overseas facilities by host country.** Analysis based on climate risk scores and facility statistics of 2233 public companies from Four Twenty Seven. The map images are created by the authors using ArcGIS. (**a**) Aggregate climate risk score, (**b**) heat stress score, (**c**) water stress score, (**d**) floods score, (**e**) sea level rise score, (**f**) hurricanes/typhoons score.

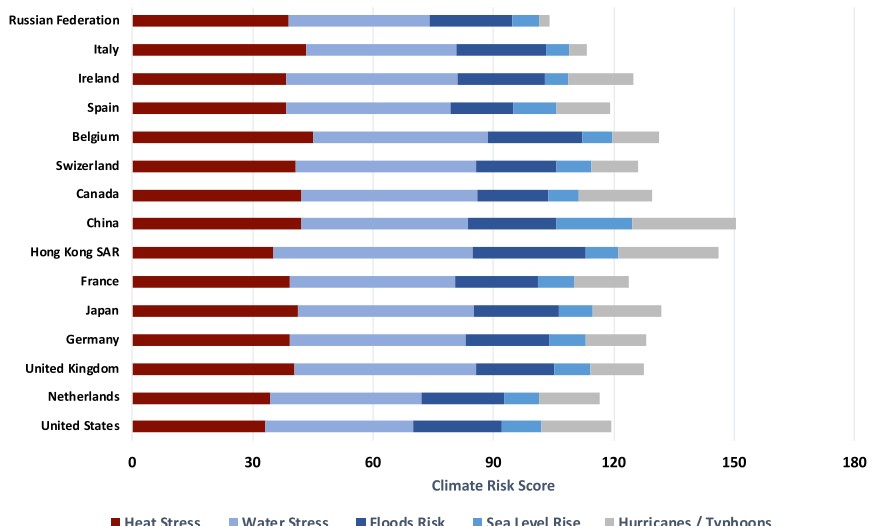

**Fig. 4 Average climate risk scores of overseas facilities by headquarters country: top 15 countries by FDI outflow stock, 1970–2019.** Analysis based on climate risk scores and facility statistics of 2233 public companies from Four Twenty Seven. FDI outflow stocks based on World Bank data.

The Supplementary Information includes robustness checks. Results are robust when we (a) remove resource-intensive industries such as mining, transportation, communications, electric, and gas (Supplementary Table 3); (b) focus on firms from the top 15 FDI exporters (Supplementary Table 4); (c) change control variables (Supplementary Table 5); and (d) aggregate climate risk drivers at the firm level (Supplementary Table 6).

We further explore why Chinese overseas facilities have higher climate risks across host countries. It may be that some Chinese companies are willing to invest in countries for political or strategic reasons, regardless of climate risks. For instance, the Belt and Road Initiative (BRI) was launched in China in 2013 to improve regional and transcontinental cooperation and connectivity through investments and trade[37]. As shown in Fig. 3, facilities in a lot of BRI countries (e.g., countries in Africa, Southeast Asia, and Latin America) face higher climate risks. Second, Chinese companies started to invest overseas aggressively in the early 2000s and may therefore have had to invest in

**Table 2 Difference of climate risks of Chinese-owned/operated overseas facilities.**

| | Model 1 - across country | | | | | Model 2 - within country | | | | |
|---|---|---|---|---|---|---|---|---|---|---|
| | Heat | Water | Floods | Sea level rise | Hurricanes/ Typhoons | Heat | Water | Floods | Sea level rise | Hurricanes/ Typhoons |
| ChineseFDI | −0.131 [0.087] | 0.149 [0.067]** | 0.273 [0.032]*** | −0.039 [0.092] | 0.503 [0.092]*** | −0.009 [0.027] | −0.088 [0.037]** | −0.074 [0.054] | −0.150 [0.097] | −0.123 [0.073] |
| Controls | | | | | | | | | | |
| GDPPerCapita | −0.078 [0.029]** | −0.047 [0.020]** | 0.010 [0.012] | 0.031 [0.013]** | −0.021 [0.016] | −0.006 [0.004] | 0.010 [0.007] | 0.004 [0.004] | 0.028 [0.008]*** | 0.002 [0.006] |
| CO2PerCapita | −0.079 [0.018]*** | −0.047 [0.027] | −0.004 [0.006] | −0.040 [0.017]** | 0.055 [0.035] | 0.000 [0.004] | −0.016 [0.017] | −0.012 [0.009] | −0.062 [0.015]*** | −0.003 [0.007] |
| Cash | 0.071 [0.187] | 0.350 [0.191] | −0.030 [0.064] | 0.340 [0.162]* | −0.591 [0.158]*** | −0.076 [0.029]** | 0.310 [0.142]* | 0.356 [0.051]* | 0.356 [0.193]* | −0.163 [0.081]* |
| Size | 0.005 [0.023] | 0.018 [0.018] | 0.020 [0.009]** | 0.050 [0.021]** | 0.026 [0.027] | −0.004 [0.005] | 0.000 [0.008] | 0.017 [0.008]* | 0.054 [0.017]*** | 0.000 [0.004] |
| ROA | −0.059 [0.585] | −0.302 [0.373] | 0.171 [0.093]* | −0.025 [0.372] | 0.299 [0.179] | −0.087 [0.151] | −0.186 [0.167] | −0.077 [0.074] | 0.137 [0.287] | 0.104 [0.044]** |
| Leverage | −0.012 [0.081] | 0.035 [0.043] | 0.011 [0.036] | 0.028 [0.052] | −0.111 [0.059]* | −0.005 [0.012] | 0.071 [0.024]** | 0.081 [0.016]*** | 0.015 [0.029] | −0.014 [0.020] |
| FirmLocalExp | 0.036 [0.015]** | −0.090 [0.019]*** | 0.015 [0.005]** | 0.024 [0.012]* | 0.090 [0.023]*** | 0.011 [0.005]* | 0.013 [0.007] | 0.006 [0.007] | 0.006 [0.010] | −0.006 [0.006] |
| Host Country FE | N | N | N | N | N | Y | Y | Y | Y | Y |
| Industry FE | Y | Y | Y | Y | Y | Y | Y | Y | Y | Y |
| N | 40761 | 40365 | 39584 | 40761 | 40761 | 40761 | 40365 | 39584 | 40761 | 40761 |
| r2 | 0.075 | 0.130 | 0.124 | 0.047 | 0.045 | 0.945 | 0.754 | 0.365 | 0.449 | 0.895 |

The unit of analysis is firm-host country-industry. Standard errors are clustered at the industry level. Outcome variables are climate risk scores and are standardized to a mean of 0 and a standard deviation of 1. GDPPerCapita, CO2PerCatpita, and FirmLocalExperience are also standardized for easy interpretation.
***$P < 0.01$; **$P < 0.05$; *$P < 0.1$.

locations with higher climate risks because the less-risky locations had already been taken[32,33]. Third, as suggested in Table 3, the climate risks of a firm's headquarter country are positively associated with those of its overseas facilities. As facilities in China have relatively high climate risks (see Fig. 3), Chinese firms are likely to take above-average climate risks when investing overseas. This is consistent with previous research suggesting that firms with local experience of high risks (e.g., natural disasters or political risks) are more likely to expand into other countries posing such risks[24,25].

## Discussion

This paper fills a research gap by assessing climate risks of FDI. We find that foreign investments have substantially and significantly lower climate risks—particularly flood, sea-level, and hurricane/typhoon—compared with all facilities across host countries. The differences of climate risks of foreign facilities are small within host countries. We also document the heterogeneities of the climate risks of overseas facilities across industries and countries. Further, we focus on China and explore whether Chinese-owned/operated overseas facilities differ from those of the global FDI. Our findings suggest that Chinese FDI have higher water, floods, and hurricanes/typhoons risks across countries, compared to all overseas facilities. Within host countries, however, the climate risks of Chinese overseas facilities are comparable with those of non-Chinese counterparts.

This study has several contributions. First, it is related to the nascent but growing literature on physical climate risks. Most recent research has focused on the financial impact of climate risks on firm performance[2,18,38], asset value[39,40], and cost of capital[19]. We expand this literature by systematically evaluating the physical climate risks of firms' FDI.

Second, the insights of this paper shed light upon the multi-disciplinary dialogue on FDI and the environment[13–15,40,41] by exploring the physical climate constraints on firms, rather than firms' environmental externalities. As firms are already being affected by climate risks, they need to add those risks into their cost function.

Third, this paper contributes to the emerging literature on Chinese overseas investment. While previous research focuses on environmental and social impacts of Chinese firms investing abroad such as carbon emissions, toxic pollutants, and ecological effects[42–45], this paper focuses on the climate risks of Chinese FDI and compares it with the global average.

Finally, our research has policy implications. Governments, investors, and communities are becoming more active in addressing their climate risks[46–51]. For instance, the Task Force on Climate-related Financial Disclosures was established in 2015 to improve and increase reporting of climate-related financial information[52]. The Network for Greening the Financial System was established in 2017 to share climate-risk–management best practices among central banks and supervisors[53]. The 2020 version of the Equator Principles incorporated climate risk assessment into its guidelines and called for climate-resilient infrastructure[54]. Understanding the climate risk baseline of firms' global assets can help policymakers and international organizations craft climate-related policies or guidelines[55–58]. For instance, the Chinese government may want to take climate risks into consideration when promoting BRI investments.

This study has limitations. First, the analysis is cross-sectional, as time-specific information on when companies built or acquired each facility was not available. Future research can collect panel data on firms' overseas projects in certain industries and examine the extent to which climate risk is a factor in choosing locations. Second, there are inherent uncertainties in climate risk data predicted by geospatial, historical, and projection models[59,60], but for now they are the best data available. Lastly, the unit of analysis is the firm-host country-industry, but for some large countries, such as the United States and China, climate risks vary within the country (e.g., coastal versus inland areas; west versus east). It would be interesting for future research to disentangle such within-country differences.

## Methods

**Data**. The assessment of firms' physical climate risks requires climate models with which to conduct forward-looking analysis, as climate risks cannot simply be calculated based on historical data. In this study, we use the physical climate risk

**Table 3 Climate risks in firms' headquarter countries and those of firms' FDI.**

| | Overseas heat | Overseas water | Overseas floods | Overseas sealevel | Overseas hurricanes |
|---|---|---|---|---|---|
| HQHeat | 0.241 [0.041]*** | | | | |
| HQWater | | 0.158 [0.024]*** | | | |
| HQFloods | | | 0.062 [0.012]*** | | |
| HQSealevel | | | | 0.102 [0.024]*** | |
| HQHurricanes | | | | | 0.222 [0.024]*** |
| GDPPerCapita | −0.021 [0.011]* | −0.017 [0.013] | 0.008 [0.010] | 0.032 [0.012]** | −0.030 [0.021] |
| CO2PerCapita | 0.012 [0.024] | −0.009 [0.032] | −0.015 [0.008]* | −0.041 [0.021]* | −0.009 [0.036] |
| Cash | 0.210 [0.175] | 0.358 [0.157]** | −0.029 [0.062] | 0.291 [0.167] | −0.599 [0.137]*** |
| Size | 0.014 [0.023] | 0.02 [0.019] | 0.022 [0.011]* | 0.048 [0.019]** | 0.029 [0.025] |
| ROA | 0.416 [0.545] | −0.074 [0.345] | 0.173 [0.085]* | −0.136 [0.387] | 0.446 [0.229]* |
| Leverage | −0.029 [0.088] | 0.054 [0.045] | −0.003 [0.050] | 0.024 [0.056] | −0.113 [0.056]* |
| Industry FE | Y | Y | Y | Y | Y |
| N | 40885 | 40488 | 39704 | 40885 | 40885 |
| r2 | 0.070 | 0.126 | 0.120 | 0.045 | 0.037 |

The unit of analysis is firm-host country-industry. Standard errors are clustered at the industry level. Outcome variables are climate risk scores and are standardized to a mean of 0 and a standard deviation of 1. *GDPPerCapita*, *CO2PerCatpita*, and *FirmLocalExperience* are also standardized for easy interpretation.
***$P < 0.01$; **$P < 0.05$; *$P < 0.1$.

scores at the firm–industry–host-country level collected from Four Twenty Seven (currently Moody's ESG Solutions). The sample covers 2233 public companies headquartered in 47 jurisdictions with more than 1 million facilities across 200 jurisdictions and 10 SIC groups. Around 28.8 percent of the facilities are outside the firm's headquarter country (i.e., overseas facilities). Facility is defined as any operational legal entity owned or operated by a company. This includes a wide range of operating activities—such as factories, offices, ports, warehouses, and stores—but does not include sites that are being developed and not yet operational. Other entities, such as European Central Bank, also use Four Twenty Seven data for climate risk analysis[61].

A facility's climate risks of its direct operations are mainly determined by the facility's location and the nature of its activities. Four Twenty Seven evaluates climate risks using several geospatial, historical, and projection models based on the specific locations of companies' facilities. The criteria for analysis include detailed climate projections that measure the change in extreme events such as heavy rainfall, high temperatures, hurricanes, coastal flooding, drought, and water stress. Four Twenty Seven's analysis focuses on extreme weather impacts (e.g., tropical cyclones) today and on other climate impacts at a mid-term projection period, 2030–2040. Supplementary Table 1 explains in greater detail the methodology, including the spatial scale, baseline period, projection period, and specific measurement for analyzing different climate risk drivers. Further, to factor the differential impacts of climate risk drivers on different economic activities, Four Twenty Seven assigns a series of sensitivity factors to the facilities that they model based on the nature of their activities. These factors vary by climate risk driver, reflecting the sensitivity and vulnerability of the company's activities to the corresponding risk factors. For example, a data center is more energy intensive than an office and, thus, will be more sensitive to the impacts of increasing temperature on energy usage. As a result, an office would receive a lower heat stress score than a data center in the same area. The Supplementary Discussion provides more details on how adjustments of climate risk scores are made based on facilities' economic activities.

Raw indicators for each climate risk driver—heat stress, water stress, floods, sea level rise, and hurricanes/typhoons—are translated into a standardized score ranging from 0 to 100; higher scores reflect higher exposure. Four Twenty Seven started to provide physical climate risk data in 2018. We use the 2019 data because it covers more public firms and facilities than the 2018 data. Also, because the evaluation of climate risk is based on the mid-term climate projection (e.g., 2030–2040) and its difference with the historical baseline, facilities' climate risk scores do not change much across years.

Like most climate projections, Four Twenty Seven's climate risk scores have limitations. First, its evaluation of future extreme weather does not necessarily

capture the most severe weather events. Second, it uses multi-model means, which may under-sample tail-end extreme events by missing processes below the model's resolution[62]. Third, there is uncertainty in modeling average shift in climate, although Four Twenty Seven applies statistical validation methods to account for model uncertainties and to ensure practicable directional accuracy.

Firm financial data are constructed from Compustat. *Size* is the natural logarithm of the book value of total assets. *Return on assets (ROA)* is the ratio of operating income before depreciation to the book value of total assets. *Leverage* is the ratio of debt (long-term debt plus short-term debt) to the book value of total assets. *Cash holding* is the ratio of cash and short-term investments to the book value of total assets. *FirmLocalExp* is a firm's average climate risk in its headquarter country, calculated from facility statistics from Four Twenty Seven. FDI outflow and inflow and country-level GDP per capita are from the World Bank. Country-level $CO_2$ emissions per capita are from Our World in Data's $CO_2$ and Greenhouse Gas Emissions database. Supplementary Table 2 reports descriptive statistics for different variables.

**Model specifications.** To assess the difference between the climate risks of overseas facilities and that of the global average across host countries, we estimate Eq. (1a) for different climate risk drivers, using the sample of all overseas and local facilities owned or operated by the 2233 public firms globally.

$$ClimateRisk_{ijc} = \alpha_j + \beta1 \times Foreign + \gamma' Controls_{ih} + \varepsilon_{ijc} \qquad (1a)$$

To assess the difference between the climate risks of overseas facilities and the global average within the same host country, we estimate Eq. (1b) for different climate risk drivers.

$$ClimateRisk_{ijc} = \alpha_j + \alpha_c + \beta2 \times Foreign + \gamma' Controls_{ih} + \varepsilon_{ijc} \qquad (1b)$$

To assess the difference of the climate risks of Chinese overseas facilities across host countries, we estimate Eq. (2a) for different climate risk drivers, using the sample of all overseas facilities owned operated by the 2233 public firms globally

$$ClimateRisk_{ijc} = \alpha_j + \beta3 \times ChinesesFDI + \gamma' Controls_{ih} + \varepsilon_{ijc} \qquad (2a)$$

To assess the difference of the climate risks of overseas facilities owned or operated by Chinese companies within countries, we estimate Eq. (2b) for different climate risk drivers:

$$ClimateRisk_{ijc} = \alpha_j + \alpha_c + \beta4 \times ChinesesFDI + \gamma' Controls_{ih} + \varepsilon_{ijc} \qquad (2b)$$

where i indexes firm, j indexes industry, c indexes host country, and h indexes headquarter country. $\alpha_j$ are industry fixed effects. $\alpha_c$ are host country-fixed effects.

$\varepsilon_{ijc}$ is the residual. The unit of analysis is firm-host country-industry. The regression is estimated by analytical weighted least squares, where the weight is the total facility count of a firm's operation in one industry and in one host country. Standard errors are clustered at the industry level.

In Eqs. (1a) and (1b), the coefficients of interest are β1 and β2, which measure the association of foreign ownership and climate risks of facilities. In Eqs. (2a) and (2b), the coefficients of interest are β3 and β4, which measure the association between Chinese ownership and climate risks of overseas facilities. Equations (1a) and (2a) include the industry-fixed effects which account for the unobserved heterogeneity of the industry. Equations (1b) and (2b) have both the industry fixed effects and host country fixed effects, which accounts for the unobserved heterogeneity of the industry and the host country.

Outcome variables are physical climate risk scores for different climate risk drivers: heat stress, water stress, floods, sea level rise, and hurricanes/typhoons. The climate risk scores are standardized to a mean of 0 and a standard deviation of 1 for easy interpretation. The inclusion of control variables mitigates the possibility that our findings are driven by some firm- or country- level omitted variables. For example, it could be that larger companies have higher climate risks; controlling for firm size and cash holdings addresses this potential confounding influence. Similarly, the other controls account for differences in performance (ROA and market-to-book), and in financing policies (leverage and cash holdings) that may correlate with a firm's investment decisions. We also control for the firm's climate risk in its headquarter country (FirmLocalExp) because firms with experience of high-impact disasters maybe more likely to expand into countries experiencing such disasters[26]. We control for GDP per capita of the headquarter country because that country's economic development level may affect firms' overseas location choices. We also control for $CO_2$ emissions per capita of the headquarter country because it may be associated with FDI and sovereign risks[15,63].

## Data availability

The data that support the findings of this study are available from Four Twenty Seven (currently Moody's ESG Solutions) but restrictions apply to the availability of these data. Data from Four Twenty Seven are proprietary and covered by Non-Disclosure Agreement, and so are not publicly available. Data are however available from the authors upon reasonable request and with permission of Four Twenty Seven.

## Code availability

The STATA code used to run the regression analysis is available from the authors upon request. Restrictions apply to the availability of the data underlying the analysis.

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

## Acknowledgements

The authors acknowledge the funding support of the ClimateWorks Foundation (19-1494, K.P.G.).

## Author contributions

X.L. conceived the study and performed the analysis K.P.G. supervised the project and oversaw the research design. X.L. and K.P.G. discussed results and edited the manuscript at all stages.

## Competing interests

The authors declare no competing interests.
