## [Peer Review File · Nature Communications]

Peer review comments –

Reviewer #1 (Remarks to the Author):

Review of Physical Climate Risk and Foreign Direct Investment: Is China Different?

The aim of Nature Communications is to publish high-quality research in all areas of the biological, health, physical, chemical and Earth sciences. The paper investigates the exposure of companies' investments to physical climate risk. As such, I believe the paper does not fit to the scope of the journal. Also by looking at other papers published by the journal, I come to the same conclusion. This impression is reinforced by the fact that the (author)s do not cite a single paper from Nature Communications.

Content wise I do not understand how the author(s) performed the analysis, which data was used for which purpose, how the data was aggregated, if the data is reliable, and how meaningful the results are. While the paper promises to investigate Foreign Direct Investment, 2/3 of the paper is about individual countries' exposure to different climate risks. Finally, I do not understand why the proposed research question is interesting for the broader audience of this journal. Let's say China is different; what are the implications? How should trade associations / Chinese businesses / other react to that? Overall, reading the paper did not convince me at all. I am sorry that I cannot be more positive.

Reviewer #2 (Remarks to the Author):

In my view, in its current form the manuscript does not appear suitable for publication in nature communications. However, it tackles an interesting and important topic and also appears to make use of a novel dataset such that the goal to make a new submission or a very fundamentally revised resubmission would seem plausible in my view. The reason is that scope and relevance appear to fit well with the journal.

To substantiate the above view, I will first provide some general comments and suggestions how – in my view – the paper might be improved and then give some more specific comments below.

General comments:

The overall message and contribution of the paper remained unclear to me. It is an interesting and to my knowledge new question, to ask if there are differences in the way how local vis-a-vis overseas facilities are exposed to or affected by climate risk.

The paper would benefit from a clear setting of which questions it will address. In this context it would be helpful to first (briefly) discuss why one may expect the climate risk exposure to be different for host versus foreign-owned facilities. This may include some general drivers of climate risk (e.g. sector/industry, location, etc.) then followed by why this should differ between those facilities which are owned by "local multinationals" or by multinationals with their headquarters somewhere else. This discussion would then serve as a good preparation to present and discuss the results (and also explain the setup in the methods section later on).

Similarly, for the case of China: what are the reasons why one may expect the case to differ (in what respect). Then (in "Results"): what does a look at the novel data tell us?

I understand that the climate risk data in all risk categories are uniquely mapped onto geo-locations. Then those geo-data is used to attach the climate risk dimensions to a facility. That may be relevant in the interpretation of drivers and results as the climate risk (in the dataset used for the analysis) is to my understanding purely determined by the location.

It often remains a bit unclear, which exact comparisons are made. It is very plausible that climate risks differ between countries (not least because the risk data are based on a location-mapping) but still interesting to look at the numbers. I understand that you are not going one step further to

argue that there is a systematic difference in the sense that some countries' firms (either abroad or "at home") are able to make some kind of a "climate-smart" location choice that is better than what other countries' firms are able to make. If I missed that, then this would in my view deserve more attention.

Whenever results or comparisons are presented, it is important to make clear and transparent what is compared with what (i.e. what changed) – see also specific comments below.

Specific comments:

Introduction, towards the end of p2 you make a strong point that so few studies focus on firms physical climate risk. My impression is different. In fact when looking at the firms reporting activities, there is way more concern with the physical climate risks as compared to, e.g., transitional risks. I would rather make that point less strong and perhaps focus more on the novelty of the foreign direct investment aspect of it, which in my view really is new.

Here some starting points where scholars have already thought about climate risks and firms. But that is NOT a call to cite all that. It is just meant FYI.

(<https://www.sciencedirect.com/science/article/pii/S0305750X20302588> ; Bernstein, Asaf, Matthew Gustafson, and Ryan Lewis, 2018, Disaster on the horizon: The price effect of sea level rise, *Journal of Financial Economics* ; Krueger, Philipp, Zacharias Sautner, and Laura T. Starks, 2018, The importance of climate risks for Institutional investors, *Switch Finance Institution research Paper No.18-58* ; awareness as early as: 2008: Corporate environmental disclosures about the effects of climate change by Elizabeth Stanny, Kirsten Ely).

On page3, still introduction, you explain why you look at the example of China. As indicated in the general comments here would be helpful to not remain vague ("Chinese overseas investments are perceived as qualitatively unique", e.g. sometimes not profit seeking etc.), but to add specific arguments that would relate to the context of FDI and climate risk.

Page 6 last sentence: "This implies that asset ownership doesn't have a strong relationship with facilities' climate risk exposure within countries." That appears important and is also plausible given the fact that location determines the climate risk in the dataset. However, this also raises and relates to the question, what the contribution and the main message of the paper is supposed to be.

Reviewer #3 (Remarks to the Author):

[see next page]

Physical Climate Risk and Foreign Direct Investment: Is China Different?

Manuscript number: **NCOMMS-21-17334-T**
Nature Communications

June 19, 2021

1 Summary

This study constructs the data, representing the physical climate risk regarding the risk exposure of overseas investments by multinational firms. To be more precise, there are two primary sectors associated with physical risk: agriculture and mining fields, before examining the single-country scenario in China. In this study, there is a shred of empirical evidence that foreign investment in China correlates with the higher level of hurricanes and typhoon risk compared to domestic activities. Meanwhile, the offshore activities of Chinese firms are likely related to the higher level of water stress, floods, and hurricanes, and typhoon risks in host countries, compared to non-Chinese firms. Eventually, this study adjusted the scale to compare between climate risks of Chinese facilities and non-Chinese ones.

2 General comments

This is an interesting topic and a well-written paper. I firmly believe that this study has some contributions to academic and practical understandings of physical climate risk. To obtain a better form of the manuscript, I would like to suggest some following points for this paper.

3 Motivation concerns:

- The 'Introduction' part should elaborate on why this study approaches the newly available data for physical climate risk. Concomitantly, the readers could benefit from knowing similarities and differences between 'physical climate risk' and 'carbon risk.'¹.
- The rationale for choosing China (due to the largest emitter) could be highlighted. Since this paper goes to an international forum, we need to understand the unique (or distinguishing) feature that China only has (No places can be found). By doing that, your study earns specific merits. Although the author(s) did the part "Is China different?", the highlighted part of China can be done in the beginning of this study.
- The results currently are just predictive power (we cannot interpret any causal relationship - since the research design does not allow us to do that). Therefore, the author could check the interpretation of your stylistic results. Concomitantly, I think China has the greatest motivation to follow the climate agreement.
- This paper employed the data in January 2020. Why did author(s) choose this data? We only capture the snapshot of climate risk scores at the firm-risk-host country-industry level in one specific year. The explanation for this choice should be done carefully.

4 Model specification concerns:

- After Equation 3-4 (Page 17-18), the author(s) should explain the set of variables rather than mentioned in the earlier paragraphs. In addition, did authors control the firm-effect as a dummy (I only found the industry-effect, or something called host country-fixed effects, and so forth)? Please explain this point carefully. In addition, Equation (3) has subscript h for the residual term ϵ . The author lacks to explain what this ϵ means. In addition, the online appendix should

¹Görge, M., Jacob, A., Nerlinger, M., Riordan, R., Rohleder, M., Wilkens, M. (2020). Carbon risk. Available at SSRN 2930897.

be acknowledged the current literature of firm characteristics, implying the possibility of predicting Climate Risk.

- China has the carbon-pilot scheme. Did you tackle this problem? In addition, some firms in the eastern coastal area actively operate in this zone. Did the author(s) control this concern? By clustering two groups (Eastern and Western coastal areas), the author(s) might see the difference (better with difference-in-difference) of firms.
- Since the literature mentioned that country risk could be stemmed from carbon emission², I also think that carbon emission in China could positively contribute to climate risk that the authors might overlook. The author could take a quick look at the current literature of forecasting the carbon emission from economic perspective in the current literature³. Therefore, putting carbon dioxide in your model could control the reverse causal bias effects.

5 Other minor points:

- In your result part, this paper will have certain merits if comparing your findings with the strand of literature.

6 Conclusion

All in all, let me conclude that this paper is very interesting. I really enjoyed reading this paper although I believe that author can do a better job.

²Chaudhry, S. M., Ahmed, R., Shafiullah, M., & Huynh, T. L. D. (2020). The impact of carbon emissions on country risk: Evidence from the G7 economies. *Journal of environmental management*, 265, 110533.

³Nguyen, D. K., Huynh, T. L. D., Nasir, M. A. (2021). Carbon emissions determinants and forecasting: Evidence from G6 countries. *Journal of Environmental Management*, 285, 111988.

Dear Nature Communications Reviewers,

We have completed a revision of our paper, “**Are Global Companies Overexposed to Climate Change? an analysis of Physical Climate Risk and Foreign Direct Investment.**”

We are very grateful for your careful reading of the paper. We have reflected a great deal on these reviews and have incorporated the suggestions into the draft we attach here. After having integrated these comments and suggestions it is now a vastly improved paper—and the first to our knowledge examining the extent to which global companies are incorporating physical climate risk into their overseas investment decisions.

The notes below provide a point-by-point response to each comment from the three referees. The original comments appear in italics, and we use bullet points to denote our responses.

Thank you again for the opportunity to be considered for publication in *Nature Communications*.

Reviewer #1 (Remarks to the Author):

The authors do not cite a single paper from Nature Communications.

- Thank you for the comment. The reason why the original paper lacked citations to Nature Communications is that we originally submitted the paper to *Nature Climate Change* but the AE thought it was a good fit in *Nature Communications* given that it was a first of its kind analysis of a phenomenon with broader appeal—global companies’ exposure to climate risk. We now cite some papers in *Nature Communications* but less than one would otherwise because the paper is the first of its kind given the new data availability discussed in our paper and the relative newness of this literature altogether. Our paper is the first to examine the physical climate risks of global companies (or more technically, ‘Foreign Direct Investment’ or ‘FDI’). We too are now convinced that Nature Communications is the right place for this analysis as it will help convey these insights to a broader audience.

Content wise I do not understand how the author(s) performed the analysis, which data was used for which purpose, how the data was aggregated, if the data is reliable, and how meaningful the results are.

- We have added significantly more explanation on the data and methods in the paper. Detail and methods about the ‘427 physical climate risk score’ are included in the Supplementary Information. 427 data is also used by other organizations and research papers in the scholarly literature. Indeed, since our last correspondence, the European Central Bank now uses 427 data for climate risk analysis (Financial Times. March 2021. ECB Stress Test Reveals Economic Impacts of Climate Change). We note those uses in the paper on page 13.
- In the method section on page 14, we also discuss the limitations of the 427 data: “Like most climate projections, Four Twenty Seven’s climate risk scores have limitations. First, its evaluation of future extreme weather does not necessarily capture the most severe weather events. Second, it uses multi-model means, which may under-sample tail-end extreme events by missing processes below the model's resolution. Third, there is uncertainty in modeling average shift in climate, although Four Twenty Seven applies statistical validation methods to account for model uncertainties and to ensure practicable directional accuracy.”

While the paper promises to investigate Foreign Direct Investment, 2/3 of the paper is about individual countries’ exposure to different climate risks. Finally, I do not understand why the proposed research question is interesting for the broader audience of this journal. Let’s say China is different; what are the implications? How should trade associations / Chinese businesses / other react to that?

- Firstly, we think this paper is seminal for the broader audience of the journal. The main point is that global firms (or FDI) has significantly and substantially lower physical climate risks, compared with the global average (including local facilities) across host countries. Within host countries, however, the differences of physical climate risks between foreign and all facilities are small. We also find that Chinese firms—which are fast becoming among the largest foreign investors—take on significantly more climate risk in overseas locations across host countries. Not only are these important academic findings but the policy implications are important. The Paris Agreement notes that all foreign investment should be aligned with Paris Goals. This paper reveals that such may not be the case with respect to physical climate risks of Chinese overseas investments.
- As suggested, we make the FDI dimension more central of the paper. We add Table 1 to compare the climate risks of facilities owned or operated by foreign companies with those

of all facilities across and within host countries. We also remove a couple of analysis on China (e.g., Foreign VS Chinese local facilities in China) so that we can focus on the main points and provide deeper analysis.

- We remove a lot of country level analysis in the first half of the paper. We still keep one map that summarizing the climate risks of facilities owned/operated by foreign firms by host country. First, it provides a landscape of FDI's regional variations in terms of physical climate risks, and we are not aware of any other paper has done that before. Second, it helps to provide one potential explanation on why climate risks of Chinese overseas investments are higher across countries. We refer to this figure again in the Mechanism section on page 11.
- In the Introduction section, we add the rationale explaining why we analyze China as a case study on page 2: "China is now among the largest outward foreign investors globally. Also, some Chinese overseas investments have political and strategic considerations (e.g., those under the Belt and Road Initiative umbrella) and are not solely profit-seeking. They may be more likely to locate in countries with higher risks (including physical climate risks) if these investments fit with the government's strategy. Further, because Chinese firms have expanded their overseas footprints only recently, they may have had to invest in locations with higher physical climate risks because the less-risky ones have already been taken." In the Discussion section, we add policy implications for China as we find higher climate risks of Chinese overseas investments (compared with global FDI) across countries.

Reviewer #2 (Remarks to the Author):

In my view, in its current form the manuscript does not appear suitable for publication in nature communications. However, it tackles an interesting and important topic and also appears to make use of a novel dataset such that the goal to make a new submission or a very fundamentally revised resubmission would seem plausible in my view. The reason is that scope and relevance appear to fit well with the journal.

- Thank you for the interest on the topic and data and constructive comments. As suggested, we have made substantial revision on the paper. For instance, we make the FDI analysis more central, and we add analysis comparing the climate risks of facilities owned or operated by foreign companies with others across and within host countries. We have also improved the writing as advised and please see the responses to the specific comments as below.

To substantiate the above view, I will first provide some general comments and suggestions how – in my view – the paper might be improved and then give some more specific comments below.

General comments:

The overall message and contribution of the paper remained unclear to me. It is an interesting and to my knowledge new question, to ask if there are differences in the way how local vis-a-vis overseas facilities are exposed to or affected by climate risk.

The paper would benefit from a clear setting of which questions it will address. In this context it would be helpful to first (briefly) discuss why one may expect the climate risk exposure to be different for host versus foreign-owned facilities. This may include some general drivers of climate risk (e.g. sector/industry, location, etc.) then followed by why this should differ between those facilities which are owned by “local multinationals” or by multinationals with their headquarters somewhere else. This discussion would then serve as a good preparation to present and discuss the results (and also explain the setup in the methods section later on).

- Thank you. As suggested, we add analysis and compare the difference of climate risks of local VS that of the global average (see Table 1 on page 6).
- Also, we discuss why one may expect the climate risk exposure to be different for host versus foreign-owned facilities in the Introduction section to serve as a preparation to present the results on page 1: “Firms considering locations in areas that are susceptible to physical climate risks will have to bear the costs of climate-related events if they occur. Firms’ decisions to locate facilities abroad involves considerations of the characteristics of the host country (e.g., market attractiveness and inputs factors) and the firms’ own capabilities. Compared with local firms, foreign firms investing abroad are at disadvantage in a local market because they face discrimination by host country stakeholders, receive more attention because they look different, and have difficulty in responding to some local conditions. To overcome the burden of foreignness and enhance their long-term competitiveness, foreign firms may be more cautious about risks in host countries, including physical climate risks. It is therefore possible that facilities owned by foreign firms have, on average, lower physical climate risks than those owned by local firms.”
- The findings are interesting in our opinion. Facilities owned or operated by foreign companies have significantly lower floods, seas level rise, and hurricanes/typhoons risks across host countries, compared with all facilities globally. While an explanation of the

causal factors that explain these patterns will need a whole new paper to address, this may be because firms are more concerned with host country risks including climate related ones when investing abroad. They may face discrimination by host country stakeholders, receive more attention because they look different, and have difficulties in responding to some local conditions. Within host countries, however, the differences are very small. This makes sense as the climate risks of facilities, whether owned by local or foreign companies, are determined by their locations and the nature of their activities (e.g., industry), and are greatly influenced by the host country's climate. Foreign companies may be less likely to invest in countries with higher climate risks, but if they do, the climate risks they face are likely to be similar to the risks of local companies.

Similarly, for the case of China: what are the reasons why one may expect the case to differ (in what respect). Then (in "Results"): what does a look at the novel data tell us? I understand that the climate risk data in all risk categories are uniquely mapped onto geo-locations. Then those geo-data is used to attach the climate risk dimensions to a facility. That may be relevant in the interpretation of drivers and results as the climate risk (in the dataset used for the analysis) is to my understanding purely determined by the location.

- Thank you. As suggested, we also add discussion on why we discuss China and why one may expect the case to differ in the Introduction on page 2: "China is now among the largest outward foreign investors globally. Also, some Chinese overseas investments have political and strategic considerations (e.g., those under the Belt and Road Initiative umbrella) and are not solely profit-seeking. They may be more likely to locate in countries with higher risks (including physical climate risks) if these investments fit with the government's strategy. Further, because Chinese firms have expanded their overseas footprints only recently, they may have had to invest in locations with higher physical climate risks because the less-risky ones have already been taken." We then discuss the results and explore potential mechanisms in the Result section.

It often remains a bit unclear, which exact comparisons are made. It is very plausible that climate risks differ between countries (not least because the risk data are based on a location-mapping) but still interesting to look at the numbers. I understand that you are not going one step further to argue that there is a systematic difference in the sense that some countries' firms (either abroad or "at home") are able to make some kind of a "climate-smart" location choice

that is better than what other countries' firms are able to make. If I missed that, then this would in my view deserve more attention.

Whenever results or comparisons are presented, it is important to make clear and transparent what is compared with what (i.e. what changed) – see also specific comments below.

- We make the comparison/writing clearer in the updated version (in Results as well as Method sections). Specifically, we move most country level comparison to the Supplementary Information but keep one map that summarizing the climate risks of facilities owned/operated by foreign firms by host country. First, it provides a landscape of FDI's regional variations in terms of physical climate risk, and we are not aware of any other paper has done that before. Second, it helps to provide one potential explanation to why climate risks of Chinese overseas investments are higher across countries, which we refer to in the Mechanism section on page 11.
- We don't discuss whether some countries' firms are able to make "climate-smart" location choice than others (we are seeking panel data within industry to tackle this interesting question in another project). What we do add in the paper is that we find climate risks of firms' headquarter countries are positively associated with that of firms' overseas facilities (see Table 3). This is consistent with previous research suggesting that firms that had local experience with high risks such as natural disasters or political risks are more likely to expand in countries experiencing these risks (Henisz and Macher, 2004; Oetzel and Oh, 2013). It also provides plausible explanation regarding why overseas facilities of Chinese firms have higher climate risks across countries.

Specific comments:

Introduction, towards the end of p2 you make a strong point that so few studies focus on firms physical climate risk. My impression is different. In fact when looking at the firms reporting activities, there is way more concern with the physical climate risks as compared to, e.g., transitional risks. I would rather make that point less strong and perhaps focus more on the novelty of the foreign direct investment aspect of it, which in my view really is new.

Here some starting points where scholars have already thought about climate risks and firms. But that is NOT a call to cite all that. It is just meant FYI.

(<https://www.sciencedirect.com/science/article/pii/S0305750X20302588> ; Bernstein, Asaf, Matthew Gustafson, and Ryan Lewis, 2018, Disaster on the horizon: The price effect of sea level rise, Journal of Financial Economics ; Krueger, Philipp, Zacharias Sautner, and Laura T.

Starks, 2018, The importance of climate risks for Institutional investors, Switch Finance Institution research Paper No.18-58; awareness as early as: 2008: Corporate environmental disclosures about the effects of climate change by Elizabeth Stanny, Kirsten Ely).

- Thank you for the suggestion and the references. We agree that the key contribution of the paper is to tackle the physical climate risks of foreign direct investment, not that few papers focused on physical climate risks. We make this point clearer in the revised version in the Introduction section.
- In the Introduction, we start with examples of climate impact of FDI and emphasize the importance. We also add one paragraph to discuss the literatures this paper speaks to and potential contribution: “Previous literature related to FDI and the environment focused on the theory of externalities, such as the extent to which firms might locate to countries that have less stringent regulation that requires firms to internalize environmental externalities; how foreign companies may spread cleaner environmental technologies or practices in host countries; or whether foreign firms have better environmental performances than indigenous firms. With respect to climate change, studies have primarily focused on the relationship between FDI and carbon emissions. While emerging literature on physical climate risk pays attention to the financial impact of climate change on firm performance, cost of capital, and asset value or price, to date there is no study on physical climate risks and FDI.”

On page 3, still introduction, you explain why you look at the example of China. As indicated in the general comments here would be helpful to not remain vague (“Chinese overseas investments are perceived as qualitatively unique”, e.g. sometimes not profit seeking etc.), but to add specific arguments that would relate to the context of FDI and climate risk.

- Thank you. We add more specific arguments that would relate to FDI and climate risk in the Introduction on page 2: “China is now among the largest outward foreign investors globally. Also, some Chinese overseas investments have political and strategic considerations (e.g., those under the Belt and Road Initiative umbrella) and are not solely profit-seeking. They may be more likely to locate in countries with higher risks (including physical climate risks) if these investments fit with the government’s strategy. Further, because Chinese firms have expanded their overseas footprints only recently, they may have had to invest in locations with higher physical climate risks because the less-risky ones have already been taken.”

Page 6 last sentence: “This implies that asset ownership doesn’t have a strong relationship with facilities’ climate risk exposure within countries.” That appears important and is also plausible given the fact that location determines the climate risk in the dataset. However, this also raises and relates to the question, what the contribution and the main message of the paper is supposed to be.

- Thank you. We agree this is a point important and we substantiate the analysis on foreign VS all facilities in the updated paper (see Table 1 and analysis).
- In the revised version we think the main messages of the paper are: 1) Globally, foreign investments have lower physical climate risks, compared with all facilities across host countries. The differences between foreign and all facilities are small within host countries. 2) We also examine China, and find that China’s overseas facilities are associated with higher water stress, floods, and hurricanes and typhoon risks across host countries, compared with all FDI. Within host countries, however, climate risks of Chinese overseas facilities are comparable to those of all overseas facilities.
- To make the messages of the paper simpler while the analysis deeper, we remove some analysis in the previous version (e.g., climate risks of foreign VS local facilities in China).
- We also clarify the contributions of the paper in the Discussion section on page 12.

Reviewer #3 (Remarks to the Author):

This is an interesting topic and a well-written paper. I firmly believe that this study has some contributions to academic and practical understandings of physical climate risk. To obtain a better form of the manuscript, I would like to suggest some following points for this paper.

- Thank you for your summary and interest in the paper!

3 Motivation concerns:

The ‘Introduction’ part should elaborate on why this study approaches the newly available data for physical climate risk. Concomitantly, the readers could benefit from knowing similarities and differences between ‘physical climate risk’ and ‘carbon risk.’¹ (IG²orgen, M., Jacob, A., Nerlinger, M., Riordan, R., Rohleder, M., Wilkens, M. (2020). Carbon risk. Available at SSRN 2930897.

- In the Introduction, we add why we approach the newly available data on page 2: “The

assessment of firms' physical climate risks requires climate models with which to conduct forward-looking analysis, as climate risks cannot simply be calculated based on historical data. In this study, we deploy newly available physical climate risk data of public companies made available from Four Twenty Seven (a Moody's affiliate)...”

- In the Introduction, we also briefly introduce the difference between “physical climate risk” and “Carbon risk”/transitional climate risk on page 2: “Note that physical climate risks are different from carbon risks or transitional climate risks - that is, risks arising from transition to a low carbon economy that affect a firms' business. A facility's physical climate risks are mainly determined by the facility's location and the nature of its activities. A facility's carbon risks are mainly determined by its energy use and technology choice.....”

The rationale for choosing China (due to the largest emitter) could be highlighted. Since this paper goes to an international forum, we need to understand the unique (or distinguishing) feature that China only has (No places can be found). By doing that, your study earns specific merits. Although the author(s) did the part “Is China different?”, the highlighted part of China can be done in the beginning of this study.

- In the introduction on Page 2, we now discuss early why China might be different: “China is now among the largest outward foreign investors globally. Also, some Chinese overseas investments have political and strategic considerations (e.g., those under the Belt and Road Initiative umbrella) and are not solely profit-seeking. They may be more likely to locate in countries with higher risks (including physical climate risks) if these investments fit with the government's strategy. Further, because Chinese firms have expanded their overseas footprints only recently, they may have had to invest in locations with higher physical climate risks because the less-risky ones have already been taken.”

The results currently are just predictive power (we cannot interpret any causal relationship - since the research design does not allow us to do that). Therefore, the author could check the interpretation of your stylistic results. Concomitantly, I think China has the greatest motivation to follow the climate agreement.

- Yes, the research design doesn't allow us to have causal interpretation. We check the interpretation of the stylistic results and other parts in the revised version to avoid the confusion. For instance, we use the word “association” instead of “cause” or “lead to”.
- Yes, China has great motivation to follow the climate agreement. Currently most climate

related agreements are focusing on carbon emission reductions or mitigation, and there are few regulations on physical climate risk and climate change adaptation. This is one of the motivations of the study – to explore whether regulations are needed to guide firms’ physical climate risk management/adaptation practice from the lens of foreign direct investment.

This paper employed the data in January 2020. Why did author(s) choose this data? We only capture the snapshot of climate risk scores at the firm-risk-host country-industry level in one specific year. The explanation for this choice should be done carefully.

- 427’s physical climate risk data is a new dataset, which started in year 2018. We can only obtain climate risk data in 2018 and 2019 when we conducted the study. We use 2019 data because it covers more public firms and facilities. Also, as the assessment of physical climate risks is based on the mid-term climate forecast (e.g., 2030-2040) and its difference with the historical baseline, facilities’ climate risk scores don’t change across years. We have made the explanation clearer in the Methodology Section on page 14.
- We agree that one limitation of the paper is that the analysis is cross sectional -- data on time specific information such as when companies build or acquire each of their individual facilities are not available. However, this is the best data we can obtain at this stage. Future research can collect panel data of firms’ overseas projects in certain industries, and we make this point clear in the Discussion section on page 13.

4 Model specification concerns:

• After Equation 3-4 (Page 17-18), the author(s) should explain the set of variables rather than mentioned in the earlier paragraphs. In addition, did authors control the firm-effect as a dummy (I only found the industry-effect, or something called host country-fixed effects, and so forth)? Please explain this point carefully. In addition, Equation (3) has subscript h for the residual term e . The author lacks to explain what this e means. In addition, the online appendix should be acknowledged the current literature of firm characteristics, implying the possibility of predicting Climate Risk.

- Thanks for the suggestion. We add the explanation of the variables in the Method section on pages 15 after the Equations.
- We don’t control for the firm-effect as a dummy because the variable of interest ChineseFDI (whether a firm is headquartered in China) would be absorbed by the firm

fixed effects. In other words, if we include the firm fixed effects, it would be collinear with the variable ChineseFDI.

- We add explanation on the residual term e . We also removed the subscript h as the headquarter country characteristics would be included in the firm characteristics i .
- We acknowledge the literature of firm characteristics and explain why we include certain control variables in the model on page 15.

China has the carbon-pilot scheme. Did you tackle this problem? In addition, some firms in the eastern coastal area actively operate in this zone. Did the author(s) control this concern? By clustering two groups (Eastern and Western coastal areas), the author(s) might see the difference (better with difference-in-difference) of firms.

- China's pilot carbon-pilot scheme is an interesting topic, but this paper doesn't tackle this problem. One reason is that we only got firms' physical climate risk at the host country and industry level. They are based on each facility's location and risk data, but we don't have geolocation data of all 1 million facilities and climate risk for each of them. As such, we couldn't conduct difference-in-difference analysis for firms or facilities operated in cities with carbon-pilot program (a large proportion in the east costal area and some in central and western China) and those without it.
- In the end of the Discussion section on page 13, we add the following: "Lastly, the unit of analysis is the firm-host country-industry, but for some large countries, such as the United States and China, climate risks vary within the country (e.g., coastal versus inland areas; west versus east). It would be interesting for future research to disentangle such within-country differences."

Since the literature mentioned that country risk could be stemmed from carbon emission², I also think that carbon emission in China could positively contribute to climate risk that the authors might overlook. The author could take a quick look at the current literature of forecasting the carbon emission from economic perspective in the current literature³.

(²Chaudhry, S. M., Ahmed, R., Shafiullah, M., & Huynh, T. L. D. (2020). The impact of carbon emissions on country risk: Evidence from the G7 economies. *Journal of environmental management*, 265, 110533. ³Nguyen, D. K., Huynh, T. L. D., Nasir, M. A. (2021). Carbon emissions determinants and forecasting: Evidence from G6 countries. *Journal of Environmental*

Management, 285, 111988.) *Therefore, putting carbon dioxide in your model could control the reverse causal bias effects.*

- Thank you for the suggestion and the references. In the revised paper, we add country level carbon emissions per capita into the model as a control variable. We also cite the paper and explain the rationale of selecting these control variables on page 15.

5 Other minor points:

In your result part, this paper will have certain merits if comparing your findings with the strand of literature.

- In the revised version, we specify the literature we speak to in the Introduction section on pages 1-2 and Discussion section on pages 12-13. Also, we add more discussion on the interpretation of the results and compare them with literature, as shown on page 5 and page 11.

6 Conclusion

All in all, let me conclude that this paper is very interesting. I really enjoyed reading this paper although I believe that author can do a better job.

- Thank you for the constructive comments. In our view, the paper has been improved as a result of your and the reviewing team's inputs.

Reviewer #1 (Remarks to the Author):

Thank you very much for sending what I agree is a substantially improved paper.

In my view it is a strength of the paper that it tackles the topic of FDI in the context of climate risk, and it also tackles that based on a relatively novel dataset.

This is why my further comments somewhat focus on transparency of the analysis and the dataset. I guess there are two particularly important points I would like to make: One is about the clarity of the econometric analysis and the second is about the climate risk dataset and how it is used in order to derive the conclusions.

As to the econometric analysis:

For me it is still a bit confusing to follow which differences are examined. Terms such as "foreign facilities" (p5), "across host countries" (p5), "difference to all facilities globally" (p3) are sometimes difficult to understand in a precise way. Maybe, a small figure or an explicit introduction of a terminology could help. I understand that in general, in a country there can be (for each industry) facilities belonging to three types of firms: (i) a local (non-multinational) firm, (ii) a local firm which is a multinational firm, (iii) the facility belongs to a multinational and non-local (foreign) firm. It sometimes is unclear which number (or average value) is referred to. But this is less severe.

To understand and interpret the econometric analysis it is important to understand the model that is estimated. Table 1 and table reflect key results. It would be helpful to be more explicit in the text what is the dependent and what is the independent variable that is estimated. If it is decided to put the explicit model equations into the appendix then that is fine, but the text should allow to understand what is done. It would also be helpful to refer to the model specification and to make it crystal clear later in the "Methods" part, which estimated equation belongs to which table. It seems eq 1a generates the results in table 1 / model 1, eq 1b generates table 1/ model 2 and so on. In my view that will make it a lot easier to really follow the analysis.

As to the climate risk dataset and how it is used:

A more important issue that needs to be reflected in my view is the following: From what I understand, the 427 / Moodys risk data that is used in the paper is used as the dependent variable as firm-specific data. However, it is unclear how that data is generated. The information as provided in the supplementary information shows how something like a location-specific risk-score is developed for the different sub-risks. However, the type of firm – or type of industry – will matter a lot for the actual risk-exposure. In fact, the authors (in my view correctly) point that out themselves towards the end of the "Introduction" on p3: "A facility's physical climate risks are mainly determined by the facility's location and the nature of its activities."

It is unclear if the data (which are a key ingredient of the paper) take some firm-specific information/data into account (and if so, how). If the "nature of its activities" is reflected in the data, this needs to be discussed as the model estimations that are applied in the paper also use a mix of firm-specific and location-specific variables as explanatory variables. It is therefore important for a meaningful interpretation of the results. If there are no firm-specific aspects ("nature of activities") reflected in the Moody's dataset, and if it is therefore a purely locationally-defined score set (meaning the risks scores are defined only by location), then this also needs to be made explicit and needs to be explicitly discussed, because then the whole discussion becomes one about firm location choice (and needs to discuss if climate risk was a driver or rather a consequence of the choice made for non-climate reasons).

Reviewer #2 (Remarks to the Author):

Dear authors,

Thank you so much for revising your manuscript according to my comments. I am delighted to accept your manuscript as the current form.

Dear Nature Communications Reviewers,

We have completed the 2nd round of revision of our paper, “**Are Global Companies Overexposed to Climate Change? an analysis of Physical Climate Risk and Foreign Direct Investment.**”

Again, we are very grateful for another round of careful reading of our paper. We have made and incorporated a number of corrections into the draft we attach here. The notes below provide a point-by-point response to each comment from the referees. The original comments appear in italics, and we use bullet points to denote our responses.

Thank you again for the opportunity to be considered for publication in *Nature Communications*.

Reviewer #1 (Remarks to the Author):

Thank you very much for sending what I agree is a substantially improved paper.

In my view it is a strength of the paper that it tackles the topic of FDI in the context of climate risk, and it also tackles that based on a relatively novel dataset.

This is why my further comments somewhat focus on transparency of the analysis and the dataset. I guess there are two particularly important points I would like to make: One is about the clarity of the econometric analysis and the second is about the climate risk dataset and how it is used in order to derive the conclusions.

- Thank you for your constructive comments, they helped us improve the paper substantially and we appreciate your guidance..
- We further revised the paper following your suggestions on the data analysis: 1) clarity of the econometric analysis; and 2) how the climate risk data is generated and used. Please see the detailed responses as below.

As to the econometric analysis:

For me it is still a bit confusing to follow which differences are examined. Terms such as “foreign facilities” (p5), “across host countries” (p5), “difference to all facilities globally” (p3) are sometimes difficult to understand in a precise way. Maybe, a small figure or an explicit

introduction of a terminology could help. I understand that in general, in a country there can be (for each industry) facilities belonging to three types of firms: (i) a local (non-multinational) firm, (ii) a local firm which is a multinational firm, (iii) the facility belongs to a multinational and non-local (foreign) firm. It sometimes is unclear which number (or average value) is referred to. But this is less severe.

- Thank you for pointing out the lack of clarity caused by different terms used in the paper. Following your suggestion, we added a figure (Figure 1 on page 5) that explains the structure of the paper and clarifies key terminologies used.
- We also tried to make these terms consistent in the manuscript. See examples on pages 4,5 and 10.

Figure 1. Paper Structure and Terminology

All Facilities	All facilities include both local and overseas facilities globally in the sample.
Local Facility (local owned / operated facility)	A facility owned or operated by either a local or multinational company whose headquarter country is the same as the host country
Overseas Facility (foreign owned / operated facility, or foreign facility)	A facility owned or operated by a multinational company whose headquarter country is different from the host country
Host Country	A country where a facility is located
Headquarter Country	A country where the center of administration of a company is located

To understand and interpret the econometric analysis it is important to understand the model that is estimated. Table 1 and table 2 reflect key results. It would be helpful to be more explicit in the text what is the dependent and what is the independent variable that is estimated. If it is decided to put the explicit model equations into the appendix then that is fine, but the text should allow to understand what is done. It would also be helpful to refer to the model specification and to make it crystal clear later in the “Methods” part, which estimated equation belongs to which table. It seems eq 1a generates the results in table 1 / model 1, eq 1b generates table 1/ model 2 and so on. In my view that will make it a lot easier to really follow the analysis.

- Thank you for your suggestions to clarify what the models estimate and what the tables present in the main text. In the revised manuscript, we explain what the outcome and explanatory variables that are estimated. We also refer to the model specification in the Methods section. For instance, we added the following text before presenting Table 1:

“The Methods section details the model specifications (Equations 1.a and 1.b) and explains the selection of control variables. We estimate Equation 1.a (Model 1) to examine whether climate risks of foreign facilities are different from those of all facilities within industry and across host countries, and estimate Equation 1.b (Model 2) to examine whether climate risks of overseas facilities are different from those of all facilities within industry and within host country. Outcome variables are physical climate risk scores for different climate risk drivers including heat stress, water stress, floods, sea level rise, and hurricanes/typhoons. The explanatory variable *Foreign* is a dummy which equals to 1 if facilities are owned or operated by foreign companies.”

(Page 5)

- We also added similar text to explain Table 2 on page 9. We keep the model specification in the Method Section to be consistent with the submission guidelines of Nature Communications.

As to the climate risk dataset and how it is used:

A more important issue that needs to be reflected in my view is the following: From what I understand, the 427 / Moody’s risk data that is used in the paper is used as the dependent variable as firm-specific data. However, it is unclear how that data is generated. The information as provided in the supplementary information shows how something like a

location-specific risk-score is developed for the different sub-risks. However, the type of firm – or type of industry – will matter a lot for the actual risk-exposure. In fact, the authors (in my view correctly) point that out themselves towards the end of the “Introduction” on p3: “A facility’s physical climate risks are mainly determined by the facility’s location and the nature of its activities.”

It is unclear if the data (which are a key ingredient of the paper) take some firm-specific information/data into account (and if so, how). If the “nature of its activities” is reflected in the data, this needs to be discussed as the model estimations that are applied in the paper also use a mix of firm-specific and location-specific variables as explanatory variables. It is therefore important for a meaningful interpretation of the results. If there are no firm-specific aspects (“nature of activities”) reflected in the Moody’s dataset, and if it is therefore a purely locationally-defined score set (meaning the risks scores are defined only by location), then this also needs to be made explicit and needs to be explicitly discussed, because then the whole discussion becomes one about firm location choice (and needs to discuss if climate risk was a driver or rather a consequence of the choice made for non-climate reasons).

- Thank you very much for the comment. Indeed, it is very important to clarify how the climate risk score is generated by Four Twenty Seven (Moody’s ESG), and whether the nature of facility activities is considered.
- Four Twenty Seven does consider both the facility location as well as its economic activities when estimating their climate risk scores. In the revised manuscript, we clarify this point in the Methods Section when explaining the data on Page 14:

“A facility’s climate risks of its direct operations are mainly determined by the facility’s location and the nature of its activities. Four Twenty Seven evaluates climate risks using several geospatial, historical, and projection models based on the specific locations of companies’ facilities. ... Further, to factor the differential impacts of climate risk drivers on different economic activities, Four Twenty Seven assigns a series of sensitivity factors to the facilities that they model based on the nature of their activities. These factors vary by climate risk driver, reflecting the sensitivity and vulnerability of the company’s activities to the corresponding risk factors. For example, a data center is more energy intensive than an office and, thus, will be more sensitive to the impacts of increasing temperature on energy usage. As a result, an office would receive a lower heat stress score than a data center in the same area.”

- We also added details on how adjustments of climate risk scores are made based on facilities' economic activities in the Supplementary Document:

“...adjustments are made based on research about the impacts of heat stress on labor as well as a Multi-Regional Input-Output (MRIO) model that informs whether certain facilities in one country are resource or water intensive than those in other countries. Water stress adjustments are based on the water demand requirements according to specific combinations of the asset's sector and country of operation. These estimates are derived from a MRIO model. Heat stress adjustments account for the energy demand requirements across different sectors according to MRIO and consider whether a sector is likely to depend upon outdoor or labor-intensive activities, which are sensitive to extreme heat conditions. The scale of the adjustments ranges from 0.75 to 1.25, providing an equal discount ($x < 1$) and penalty ($x > 1$) of 0.25 to those facilities that fall above or below average ($x = 1$) for all sectors.”

- We reviewed the interpretation of the results and made some adjustments to ensure its accuracy. See examples on pages 6 and 10.

Reviewer #2 (Remarks to the Author):

Dear authors,

Thank you so much for revising your manuscript according to my comments. I am delighted to accept your manuscript as the current form.

- Thank you very much again for your constructive comments, which helped us to improve the paper.

Peer review comments, third round - –

Reviewer #1 (Remarks to the Author):

Dear author(s)! Thank you very much for the revised manuscript. In my view the paper improved a lot. The changes that you made, do in my view substantially improve the transparency of what has been done and therefore support the main contribution of the paper: analysing FDI in a climate risk context and applying this relatively novel dataset.
I am happy to suggest to accept the paper.

Dear Nature Communications Reviewers,

We have completed the final revision of our paper, “**Assessing the Climate Change Exposure of Foreign Direct Investment.**” (NCOMMS-21-17334B)

Again, we are very grateful for another round of careful reading of our paper. The notes below provide a point-by-point response to each comment from the referees.

Thank you again for the opportunity to be considered for publication in *Nature Communications*.

Reviewer #1 (Remarks to the Author):

Dear author(s)! Thank you very much for the revised manuscript. In my view the paper improved a lot. The changes that you made, do in my view substantially improve the transparency of what has been done and therefore support the main contribution of the paper: analysing FDI in a climate risk context and applying this relatively novel dataset.

I am happy to suggest to accept the paper.

- Thank you very much again for your constructive comments, which helped us to improve the paper.